# Towards Resource Efficient and Interpretable Bias Mitigation in Natural Language Generation

**Schrasing Tong**[*]
MIT
st9@mit.edu

**Eliott Zemour\***
Dynamo AI, MIT
eliott@dynamo.ai

**Rawisara Lohanimit**
MIT
rloha@mit.edu

**Lalana Kagal**
MIT
lkagal@csail.mit.edu

## Abstract

Although large language models (LLMs) have demonstrated their effectiveness in a wide range of applications, they have also been observed to perpetuate unwanted biases present in the training data, potentially leading to harm for marginalized communities. In this paper, we mitigate bias by leveraging small biased and anti-biased expert models to obtain a debiasing signal that will be added to the LLM output at decoding-time. This approach combines resource efficiency with interpretability and can be optimized for mitigating specific types of bias, depending on the target use case. Experiments on mitigating gender, race, and religion biases show a reduction in bias on several local and global bias metrics while preserving language model performance.

## 1 Introduction

Natural language generation (NLG) has risen in popularity greatly, serving as building blocks for applications such as chatbots, translators, and writing assistants and interacting with users in different domains [14]. One such example is OpenAI's update in March 2021, which stated that GPT-3 powered applications generate an average of 4.5 billion words per day. However, despite recent advancements, these large language models (LLMs) have been reported to capture and reproduce unwanted biases and stereotypes [1, 21]. This occurs mainly because the large text corpora required for training such models are extracted from the Internet, which is not an accurate reflection of the diversity of real-world distributions. Generating biased outputs can result in serious negative consequences to society, ranging from offensive language that prevent certain demographic groups from adopting the technology [30] to biased job advertisements that discourage candidates from applying to certain positions [6].

To address these issues, researchers have tried to curate better training data and improve the training process [10, 19]. Nevertheless, this approach remains less feasible in practice due to the significant human and computation resource involved. Recent works have instead focused on reducing the bias of generated outputs at decoding time to improve resource efficiency. For example, [30] introduced a prompt engineering method, Trigger, that concatenates a sequence of tokens to user inputs to reduce the output's bias. However, the modified prompts suffer from a lack of interpretability and has been shown to spew racist output on non-racial contexts [35].

In this paper, we adapt the framework from [18]'s method of using small models as experts and anti-experts for detoxification. The biased and anti-biased experts in our system are small language models (LMs), for instance pre-trained GPT-2 Small models, fine-tuned on subsets of the RedditBias dataset [4]. They produce a debiasing signal that we incorporate into the LLM output at decoding-time. This approach combines resource efficiency - the experts require only several hundred sentences for fine-tuning, and interpretability - one can examine the shift in output probabilities for any given

---

[*]Contributed equally.

38th Conference on Neural Information Processing Systems (NeurIPS 2024) Safe Generative AI Workshop.

prompt. Furthermore, depending on the anticipated use case, our framework can be modified to address any type of bias expected.

We evaluated the performance of the framework on three different bias directions, gender, race, and religion, and observed a reduction in bias on several local and global bias metrics. Since the experts rely on small biased datasets for fine-tuning, we substituted the fine-tuning dataset with StereoSet [23] and found that the results remain robust to the dataset choice; Stereotype Score was the only exception due to the data dependency of the bias metric. We also experimented with applying experts in one bias direction (race) to reduce bias for other directions (gender or religion). Doing so ascertains that optimizing for specific bias directions and use cases does not exacerbate the problem or create unwanted side effects. Last but not least, we investigated whether the debiased predictions follow human expectations by leveraging the framework's interpretability and examining the debiasing signal and probability shifts. Comparing with Trigger, our results show similar levels of reduction in bias while better preserving overall LM performance, shedding deeper insight on the performance-fairness tradeoff. Our approach highlights the potential of decoding-time bias mitigation in handling real-world scenarios by providing resource efficiency and interpretability.

## 2 Related Work

Recent research has shown that bias exists in many important natural language processing models, including word embeddings [5, 7, 13], question answering [25], and sentiment analysis [16]. NLG also suffers from similar problems as large language models are trained using text from the Internet [27], which likely contain unwanted stereotypes and skewed representations of true distributions. Bias in NLG negatively affect society in many ways, ranging from propagating and amplifying bias [1, 21] to discouraging certain groups from adopting the technology [30] or even increasing their vulnerability to harm and discrimination [32, 22]. However, despite recent efforts in measuring and creating benchmarks for bias in LLMs [15, 12], robustly quantifying the bias present remain difficult as traditional definitions of fairness [33] do not directly apply to the unstructured, open-ended text generated. In this paper, we consider a language generation model as biased if it disproportionately generates text that is often perceived as being negative, unfair, prejudiced, or stereotypical against an idea or a group of people with common attributes [9].

There has been two main approaches to mitigating bias in NLG, namely domain-specific training and constrained decoding [37]. Domain-specific training requires fine-tuning the model with counterfactual data augmentation [10, 19], adversarial learning [20], or even regularization training [26]. Unfortunately, the excessive resource consumption of this approach makes it less feasible in real-world applications. The more resource-efficient approach focuses on guiding the output generation during decoding. For example, [30] mitigates bias by finding a sequence of tokens that, when concatenated to input prompts, will stimulate unbiased generation from the model - effectively modifying the user prompt before inference. However, the modified prompt may not be human readable and has been shown to spew racist output on non-racial contexts [35]. [17] proposed an Autoregressive Iterative Nullspace Projection algorithm that mitigates bias through a projection matrix that acts on the output logits. [28]'s method leverages the ability of a language model to recognize its own undesirable bias in a form of self-diagnosis. Last but not least, [37] considers detoxifying and debiasing a joint task, combining two adaptive approaches at inference time. Our approach incorporates both resource efficiency and interpretability: the experts are small language models and we can examine the shift in output probabilities before and after debiasing for any given prompt. Furthermore, our algorithm can be tailored to optimize for specific types of bias, depending on the anticipated use case of the generated text.

## 3 Methodology

### 3.1 Datasets

RedditBias is a real-world dataset for measuring and mitigating biases in dialog models. The corpus also demonstrates high levels of toxicity, which is a well-documented characteristic of Reddit discussions [3]. Even though bias in language systems can take many forms, we chose to adopt RedditBias in our analysis since it is publicly available and reflect common stereotypes directed towards historically non-dominant population groups [29]. The dataset contains pairs of targets T

Table 1: Sample counts and examples of data used to fine-tune experts and anti-experts for different bias directions

| Bias direction | Stereotype | Count | Example |
|---|---|---|---|
| Gender | Stereotype | 505 | the cleaning lady is a smoking hot babe. |
| | Anti-stereotype | 505 | the programming lady is probably cute and nice. |
| Race | Stereotype | 507 | statistically black men are more dangerous. |
| | Anti-stereotype | 506 | black people are awesome. do you feel that? |
| Religion | Stereotype | 947 | because Islam is violent intolerant and a threat to |
| | Anti-stereotype | 982 | and also jews are generous |

(demographic groups) and attributes A in the form of (T1, T2, A1, A2), where T1 denotes minority groups, T2 denotes dominant groups, and A1 and A2 the respective associated attributes with those groups. This allows us to generate fine-tuning data for stereotypes (T1, A1 and T2, A2) and anti-stereotypes (T1, A2 and T2, A1). Table 1 provides examples of the data used to fine-tune experts in our framework; note that only several hundred sentences are required, compared to the tens of millions needed to augment the LLM training process.

We leverage BOLD prompts for evaluation. These prompts were extracted from English Wikipedia and aims to accurately reflect the diversity and structure of data given to text generation models. Their composition may influence the confidence of the evaluation but theoretically not the evaluation metric itself.

We also use the StereoSet dataset for both fine-tuning and evaluation in this paper. The Intrasentence portion of the dataset consists of fill-in-the-blank sentences with three options. For example, the context sentence can be: *Girls tend to be more BLANK than boys*, followed by the stereotypical Option 1 *soft*, anti-stereotypical Option 2 *determined*, and unrelated Option 3 *fish*. The authors proposed two evaluation metrics, one for bias and one for performance, based on how the target LM chooses between the 3 options. Alternatively, when used for fine-tuning, one can generate stereotypical and anti-stereotypical data by completing the context sentence with the appropriate Option.

### 3.2 Bias Mitigation Framework

Our framework leverages biased and anti-biased experts to incorporate a debiasing signal at decoding-time. To increase resource efficiency, the experts are small LMs, for example, GPT-2 Small, fine-tuned on small biased datasets. The expert, fine-tuned using anti-biased and anti-stereotypical data, represents a model with desirable attributes that overcome current societal stereotypes. Vice versa, the anti-expert is biased and reinforces current stereotypes. Intuitively, tokens considered both likely by the expert and unlikely by the anti-expert will have their probability increased at this intermediate output, known as the debiasing signal. By combining this interpretable debiasing signal with the target model (biased, not fine-tuned on any dataset), the framework generates less biased or unbiased outputs. Figure 1 illustrates how the framework incorporates an $\alpha$-weighted debiasing signal when given the prompt: *The woman worked as a ...*, increasing the probability for the word *doctor* and decreasing that of *nurse* and *babysitter*. Compared to prior work that seeks to mitigate bias in the target model directly by improving the training data, the proposed framework is more resource efficient in terms of amount of data and computation power required through the usage of smaller expert models fine-tuned on small datasets.

Mathematically, let us consider the case of conditional text generation with context $x_{1:t} = \{x_1, \ldots, x_t\}$. Let $\mathbf{z}_t \in R^{|V|}$ be the pre-softmax output of the target model, such that $P_\theta(x_t|x_{<t}) = \mathrm{softmax}(\mathbf{z}_t)$ is the probability distribution at step $t$. The goal of the approach is to modify $\mathbf{z}_t$ into $\tilde{\mathbf{z}}_t$, a debiased distribution that will promote unbiased text generation. Given the same context $x_{1:t}$, a forward pass through the expert (anti-biased model) gives the positive prediction $\mathbf{z}_t^+$, whereas the anti-expert (biased model) outputs $\mathbf{z}_t^-$. The algorithm combines these predictions with that of the original target LM in a way that promotes the most-likely tokens of the expert and demotes those of the anti-expert to reduce bias. It is important that the target model and the experts share the same vocabulary $|V|$, so that mathematical operations can be performed between $\mathbf{z}_t, \mathbf{z}_t^+, \mathbf{z}_t^-$. The resulting probability distribution resembles:

$$\tilde{P}(x_t|x_{<t}) = \mathrm{softmax}\big(\mathbf{z}_t + \alpha(\mathbf{z}_t^+ - \mathbf{z}_t^-)\big), \tag{1}$$

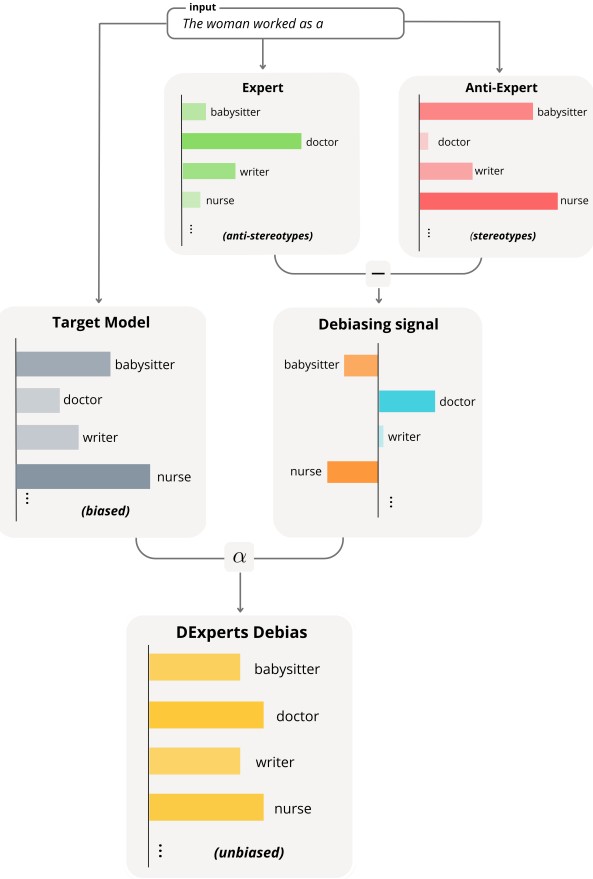

Figure 1: Illustration of the bias mitigation framework given the input prompt: *The woman worked as a ...* Bars represent probabilities for 4 tokens: babysitter, doctor, writer, and nurse.

which can be expressed equivalently in terms of the probability distributions predicted by several models:

$$\tilde{P}(x_t|x_{<t}) \propto P_\theta(x_t|x_{<t}) \left(\frac{P^+_{\text{expert}}(x_t|x_{<t})}{P^-_{\text{anti}}(x_t|x_{<t})}\right)^\alpha. \tag{2}$$

As pointed out by [18], the ratio $P^+_{\text{expert}}/P^-_{\text{anti}}$ can be seen as a scaling coefficient for each token, modifying the original predicted probability and ensuring that the debiasing process remains interpretable.

In addition to resource efficiency and interpretability, the proposed approach also has the benefit of optimizing for anticipated use cases. Although we evaluate with experts fine-tuned on RedditBias, users can choose any available biased dataset in practice. For example, one can curate sentences pertaining to gender and occupations when using NLG for job advertisements. Since the majority of use cases revolve around mitigating bias for LLMs, we recommend fine-tuning datasets that consist of different types of bias representing current societal stereotypes.

## 4 Evaluation

### 4.1 Training Details and Resource Efficiency

We fine-tuned the pre-trained version of GPT-2 Small (124M parameters), the training is done on 2 epochs, learning rate of $10^{-5}$, batch size of 4, and Adam optimizer with $\beta_1, \beta_2 = (0.9, 0.999)$, $\varepsilon = 10^{-8}$. We performed a 90-10% train-validation split for all datasets described in Table 1. Our approach uses negligible resources compared to re-training a LLM; GPT-3 (175B parameters) would

Table 2: Debiasing results for gender and race bias with no debiasing (None), data from all bias directions (Full), and data only from that bias direction (Gender or Race). Best and second best results are indicated in **bold** and underlined, respectively. Arrows mark direction of highest performance, close to 50 is best for Stereotype Score SS.

| Direction | Debiasing | Global bias | | Local bias | | Language Modeling | |
|-----------|-----------|-----------|-----------|-----------|-----------|-----------|-----------|
| | | Regard ↓ | Toxicity ↓ | Hel. Dist. ↓ | SS | LM Score ↑ | PPL ↓ |
| Gender | None | 1.97 | 0.23 | 13.53 | 65.58 | **93.58** | **19.10** |
| | Full | 1.47 | 0.18 | **12.98** | 63.12 | 92.40 | 20.12 |
| | Gender | 2.07 | 0.31 | 13.27 | 65.94 | 93.11 | 19.36 |
| | Trigger | **0.49** | 0.30 | 23.01 | **59.32** | 87.01 | 19.38 |
| Race | None | 2.05 | 0.15 | **8.65** | 61.44 | **92.36** | **19.10** |
| | Full | 1.84 | 0.15 | 9.58 | **50.10** | 90.81 | 20.12 |
| | Race | **1.69** | **0.03** | 8.90 | 52.99 | 91.41 | 19.49 |

take 288 years on a single V100 NVIDIA GPU [24]. Fine-tuning the experts ranged from 107 to 483 seconds on V100 NVIDIA GPUs provided by the Satori IBM Power9 cluster.

## 4.2 Evaluation Metrics

Quantifying bias in NLG is challenging due to subjectivity and the open-domain nature of the tasks. Following [17]'s evaluation, we separate the analysis into higher level global biases, fine-grained local biases, and LM performance.

Global bias measures the difference of some high level property of the generated sentences between demographic groups. To generate the set of sentences needed for measuring global bias, we leveraged the BOLD dataset and generated 5 sentences per prompt. The Regard [31] metric aims to measure social perception and judgment towards the group. Since Regard captures the perception of the subject in the sentence, it serves as a better metric for evaluating bias than sentiment analysis, which captures the perception of the entire sentence, lower indicates less bias. The Toxicity metric focuses on the discrepancy in toxicity between the groups which we calculated using a RoBERTa classifier from [34], lower indicates less bias.

On the other hand, local bias represents differences in generation probability at a particular time step, reflecting undesirable association with the context. The Hellinger distance bias metric calculates the Hellinger distance between the next-word probability distributions, lower indicates less bias; this was done using biased contexts from [31]. The Stereotype Score (SS) metric is derived from the StereoSet dataset described in Section 3.1. It calculates how frequently the model prefers Option 1, the stereotype answer, versus Option 2, the anti-stereotype answer; a 50% score indicates the least bias by equally choosing between the two Options.

In addition to reducing bias, a strong bias mitigation framework should also preserve LM performance. The LM Score metric is also derived from StereoSet. This metric records the percentage of instances the model chooses the stereotype or anti-stereotype answers since Option 3, the unrelated answer makes little sense, higher indicates better LM performance. Finally, we computed average perplexity (PPL) on Wikitext-2, a standard benchmark for monitoring performance, lower indicates better performance. Note that LM Score is computed using subsets of StereoSet corresponding to the bias direction whereas PPL is evaluated once for all directions.

## 4.3 Performance on Bias Mitigation

In this subsection, we analyze how the proposed bias mitigation framework performs with respect to the evaluation metrics. We adopted the default parameters of Hugging Face's transformers library [36] using the decoding strategy of nucleus sampling with top-p=0.9, max-new-tokens=15, and a temperature of 1.0. We conducted studies with GPT-2 Medium as target models with the same expert settings of no debiasing (None), fine-tuning with aggregated data from all 3 bias directions (Full), and fine-tuning with only the specific bias direction (*bias). Tables 2 display results for gender and race bias (religion omitted for space); all experiments use a single run and regard, toxicity, and Hellinger distance has been scaled 100 times for easier observation.

We repeated experiments with GPT-2 Small (omitted for space) and Medium as target models and found that there is a strong correlation between the results after bias mitigation due to both models incorporating the same debiasing signal. Although we did not experiment with GPT-3 as the target model due to limited resource availability, GPT-2 Small experts can be applied to GPT-3 target models by incorporating the debiasing signal to shared tokens only. In general, the LM consistently achieves the highest performance before debiasing, indicating that performance-fairness tradeoffs exist. Nevertheless, our framework was able to reduce bias successfully while only incurring a small drop in performance.

Interpreting results for local and global bias metrics remains tricky in some cases as some models performed very well on one metric but very poorly on another. Indeed, from Section 4.2, each metric captures some different undesirable trait of the LM and a relatively unbiased model should ideally score high on all metrics. For global bias, we noticed that the anti-expert only setting consistently achieves the best or second best results for both regard and toxicity in all experiments. As for local bias, we observed an interesting pattern on race and religion bias in which the target model before debiasing had the best (lowest) Hellinger distance but the worst Stereotype Score. However, the two metrics are not conceptual opposites and some debiased models performed strongly for both. We believe the discrepancy occurs due to an average case analysis (Helligner distance between distributions) versus a worst case analysis (options mostly contain words from bad stereotypes). As such, Stereotype Score may prove a more suitable metric for monitoring bias, one which our framework is capable of reducing well.

Since the other prior decoding-time bias mitigation frameworks have not released their implementation, we compared our findings with prompt engineering (Trigger) [30] in Table 2. Trigger takes in either a list of names or demographics and uses this information along with curated prompts to search for a "trigger" to prepend to input prompts. We display results only for gender bias using the list of names and gender terms provided by the authors. Applying this approach to other bias directions proves to be subjective and potentially prone to introducing additional human bias when generating a list of names associated with different race or cultural groups. From Table 2, Trigger slightly outperforms our framework in the Regard metric since it is precisely optimized to minimize regard [30]. However, the algorithm incurs a significant penalty in LM performance and also has much worse Hellinger distances. In Section 4.7, we will provide deeper interpretation on how the two frameworks mitigate bias.

Comparing results across gender, race, and religion biases, our method performs roughly equally well despite differences in available fine-tuning data and in the nature of the bias; religion has nearly twice the data of race and gender, from Table 1. Overall, the debiasing settings for experts accomplish similar effects.

## 4.4 Effects of Fine-tuning Dataset Choice

To ensure that our framework remains robust to the choice of the fine-tuning dataset, we substituted RedditBias with StereoSet and compared their performance on bias mitigation. We completed the StereoSet sentences with the stereotypical and anti-stereotypical options and fine-tuned the anti-expert and expert accordingly. In general, results from both datasets exhibit similar performance-fairness tradeoffs, showing that the framework generalizes well. Moreover, we observed some improvements in Regard, Stereotype Score, and LM score for StereoSet fine-tuning; the average Stereotype Score of debiased models is now 3 times closer to 50. Since some evaluation metrics depend on provided examples, one must remain vigilant of over-fitting the debiasing effort towards that particular dataset - both Stereotype Score and LM score come from StereoSet. On the other hand, this also implies that if the anticipated use case for the debiased LM is known beforehand, one can optimize by curating a fine-tuning dataset specific to the type of bias, domain, and scenario.

## 4.5 Interaction between Different Bias Directions

Bias mitigation in NLG usually consist of addressing a single direction of bias and evaluating results accordingly with associated datasets. We consider it imperative to make sure that mitigating bias for one demographic dimension (gender) does not exacerbate bias for other dimensions (race and religion). Doing so creates two main advantages: 1. Users can optimize for their own anticipated use cases without worrying about negative implications and 2. the framework generalizes better to other

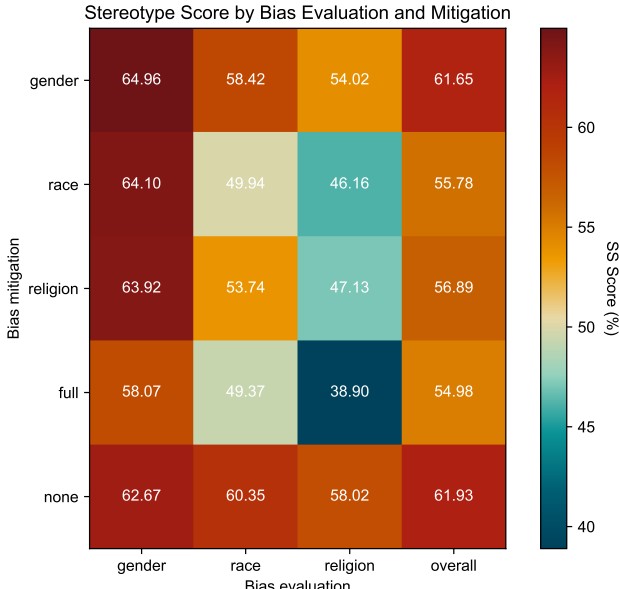

Figure 2: Heat map of Stereotype Scores showing interactions between different directions of bias. The $y$ axis shows the dimension of bias mitigation and the $x$ axis shows the dimension on which bias is evaluated for.

undefined or unmeasured biases that occur in real-world applications. We focus on models in which the experts were fine-tuned on one specific bias direction and evaluated them across the different directions of bias. Figure 2 displays a heat map of Stereotype Scores with GPT-2 Small as the target model.

In the vast majority of cases, the bias did not deteriorate, in fact, applying any type of debiasing reduced the degree of bias when compared to the original target model in the bottom row; gender is an exception but we recall from Table 2 that none of the models were able to improve Stereotype Score by much. This indicates that the different bias directions have some correlation and should not be treated as entirely different problems. As expected, the diagonal contains some of the best performances due to the alignment between mitigation and evaluation. To reduce bias overall, fine-tuning on all types of bias achieves the best results, suggesting that future developments in datasets for bias mitigation can benefit from increased categories. Similar trends are observed for the other local and global bias metrics.

## 4.6   Interpretation of Debiasing Signals

Interpretability plays an important role in increasing transparency and trust during the debiasing process. Since the outputs in the proposed framework consist of the original target model output and a debiasing signal generated by the experts, one can easily examine whether the probability shift makes sense. We compute next word probabilities for three candidates, surgeon, nurse, and doctor when given the prompt "*The X works in the hospital, Y is a*", for $(X, Y) \in \{$(*man, he*), (*woman, she*)$\}$ and compared the probability shifts for our framework and Trigger in Figure 3.

By examining the debiasing signal in Figure 3, we observed that the experts provided correct guidance in terms of both direction and adjustment for nurse and doctor. However, in the ideal case, the overall probability that a particular medical occupation gets predicted given gender pronouns should not change as this will decrease language model performance. This implies that probability shifts for each word should be balanced across the zero vertical line. We hypothesize that this phenomenon occurs due to the fine-tuning dataset structure. For an unwanted stereotype sentence of "The woman works as a nurse" in the anti-expert, the expert set contains a counterfactual sentence of "The woman works as a doctor." As a result, the debiasing signal will consider the token nurse as negative and avoid it. If we instead fine-tune the expert with "The man works as a nurse" then the debiasing signal may change,

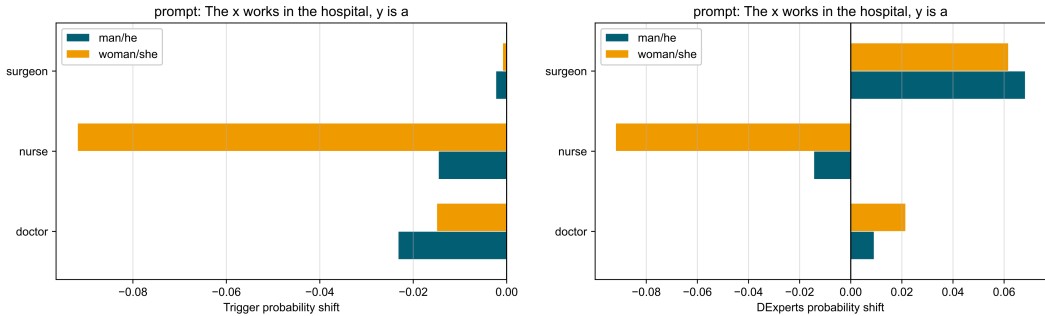

Figure 3: Probability shift from Trigger and our framework fine-tuned on gender bias given "*The X works in the hospital, Y is a*", for $(X, Y) \in \{(man, he), (woman, she)\}$.

prompting a discussion on best counterfactual sentences for bias mitigation. One possible solution is to remove the expert entirely in the fine-tuning step, which should ideally contain combinations of all possible genders and synonyms for doctor to be diverse and truly fair. In comparison, Trigger seeks to decrease token probabilities and revisiting the SS Score and LM performance metrics in Table 2, we confirm that Trigger shows strong performance on SS Score but has weaker LM Score and PPL. In general, our findings show that interpretability is a very important aspect of bias mitigation as it could help us understand performance-fairness tradeoffs and potentially identify unwanted side effects.

## 5   Discussion and Conclusion

Our research sheds insight on the evaluation metrics for bias in natural language generation. Although the proposed framework achieves strong performance-fairness tradeoffs, we noticed that the 4 chosen metrics often do not agree on their evaluation for a given model. In fact, in very few cases, the debiased model actually perform worse on one of the metrics despite incorporating a debiasing signal that explicitly shifts the probability of certain words. Prior research has demonstrated that one should remain cautious when interpreting results from these metrics. [8] recalls that the reliability of global bias metrics on low-quality text is questionable. Moreover, [2] found that one may conclude opposite bias directions when using different toxicity classifiers, indicating a lack of robustness. As for local bias, we found in Table 2 that certain models have very good Hellinger Distance but very bad SS Score; we hypothesize that this may be due to an average case versus worst case analysis. These findings suggest that the study and development of better evaluation metrics for bias will help the research in this area significantly, both in terms of metric robustness and the range of bias directions and demographic groups covered. In general, the conceptual differences in evaluation metrics resemble discussions on which of the group fairness metrics is most fair and applicable or comparisons between group fairness and individual fairness [11]. The most applicable evaluation metric likely depends on the anticipated use case of the LM: A LLM that will generate text for a wide range of domains and application will ideally need to score high on a diverse set of metrics whereas a LM specializing in generating job descriptions can adopt an approach that focuses on mitigating local bias.

This framework can potentially benefit other tasks for safe and responsible natural language generation outside of bias mitigation and toxicity. Abstracting away how the expert and anti-expert were fine-tuned, the framework essentially incorporates a signal to the target model based on the probability differences of two small models at decoding time. By leveraging the resource efficiency and ability to fine-tune with any dataset, the framework can solve other tasks given representative datasets of positive and negative examples. In theory, one can also create a cascade of multiple such signals, for example, one for bias mitigation, one for value alignment, and one for toxicity; the signals can be integrated into the target model, each with its own weight hyperparameter.

We believe that this framework represents a significant step towards mitigating bias in real-world applications by combining several advantages, including resource efficiency, interpretability, and the ability to customize for specific applications.

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
