# OpenReview forum: "Towards Resource Efficient and Interpretable Bias Mitigation in Natural Language Generation"
_NeurIPS.cc/2024/Workshop/SafeGenAi — SafeGenAi Poster_

### Official Review · Reviewer_5bpV · 2024-10-08
**A review of Towards Resource Efficient and Interpretable Bias Mitigation in Natural Language Generation**

**Rating:** 6
**Confidence:** 4

**Review:**

Summary: This work uses small bias expert models and anti-bias expert models to add debiasing signals in the decoding process, thereby reducing multiple biases such as gender, race, and religion. Compared with previous bias mitigation methods that require a lot of data and computing resources, this method is more resource-efficient and interpretable.

**Strengths**:

1. This work trains biased and anti-biased experts using Stereotype and Anti-stereotype datasets, generating a debiasing signal that alters the distribution of words during inference to reduce stereotypical bias. The approach is highly **innovative**.
2. By leveraging smaller base models, it achieves bias mitigation with **fewer computing resources**.

**Weaknesses**:

1. If the proposed method is applied to target models that are **black-box APIs**, where token probability distributions are not accessible, how can this pipeline be used to reduce bias?
2. How does this work handle **out-of-distribution** cases that are not covered in the training dataset?